# Mechanical Properties of Ti_3_AlC_2_/Cu Composites Reinforced by MAX Phase Chemical Copper Plating

**DOI:** 10.3390/nano14050418

**Published:** 2024-02-24

**Authors:** Cong Chen, Zhenjie Zhai, Changfei Sun, Zhe Wang, Denghui Li

**Affiliations:** 1School of Physics and Electronic Information Engineering, Qinghai Minzu University, Xining 810007, China; 15562091462@163.com (Z.Z.); changfeisun@163.com (C.S.); z19917306057@163.com (Z.W.); li15039538994@163.com (D.L.); 2Qinghai Key Laboratory of Nanomaterials and Technology, Xining 810007, China

**Keywords:** copper, Ti_3_AlC_2_, composites, compression strength

## Abstract

Among the various reinforcement phases available in Cu-based composites, the unique layered structure and easy diffusion of A-layer atoms make MAX phases more suitable for reinforcing a copper matrix than others. In this study, Cu-coated Ti_3_AlC_2_ particles (Cu@Ti_3_AlC_2_) were prepared through electroless plating, and Cu@Ti_3_AlC_2_/Cu composites were fabricated via vacuum hot-press sintering. The phase composition and microstructure of both Cu@Ti_3_AlC_2_ powder and composites were characterized using X-ray diffraction (XRD), scanning electron microscopy (SEM), and transmission electron microscopy (TEM). The results demonstrate the creation of successful electroless copper plating to obtain a Cu coating on Ti_3_AlC_2_ particles. At 850 °C, a small amount of Ti_3_AlC_2_ particles decompose to form TiCx, while Al atoms from the A layer of MAX phase diffuse into the Cu matrix to form a solid solution with Cu(Al). The test results reveal that the density of the Cu@Ti_3_AlC_2_/Cu composite reaches 98.5%, with a maximum compressive strength of 705 MPa, which is 8.29% higher than that of the Ti_3_AlC_2_/Cu composite. Additionally, the compressive strain reaches 37.6%, representing an increase of 12.24% compared to that exhibited by the Ti_3_AlC_2_/Cu composite.

## 1. Introduction

The general formula of a MAX phase is M_n+2_A_n_X_n+1_, where M is an excess metal element; A is an element of the A main group (from the third main group to the sixth main group); and X is an element of carbon or nitrogen. As a ternary carbide or nitride phase with a nanolamellar structure, a MAX phase usually has both high strength and good thermal shock and oxidation resistance like ceramics, as well as high electrical and thermal conductivity like metals [1]. In addition, MAX phase materials have properties such as low density and a high modulus [2].

Studies have shown that MAX phase materials have a unique mechanical behavior at high temperatures, such as the outward diffusion of atoms in the A-layer, which is attributed to their microscopic, layered structure and decomposition properties [3]. This property facilitates the formation of good interfacial bonding between phases on the one hand and excellent self-healing and oxidation resistance on the other hand [3]. Therefore, MAX phases are considered to be very promising reinforcing phases for metal matrix composites. However, the imbalance between a MAX phase and metal matrix (Cu, Fe, Ni, etc.) at high temperatures makes the preparation of MAX-phase-reinforced metal matrix composites difficult [4,5,6,7].

Cu has been widely used in the field of electronics and electrical appliances due to its excellent electrical and thermal conductivity and very high ductility, but its mechanical properties also impose a great limitation to its own application [8]. To address this deficiency, current solutions mainly involve the preparation of alloys and the addition of a second reinforcing phase to create copper matrix composites (CMCs). Compared to CMCs, alloys have the disadvantage of being prone to precipitation roughening, leading to a loss of strength [9]. Therefore, CMCs have received much attention from researchers and are considered to be highly promising materials for advanced electrical applications [10]. The choice of reinforcing phase is extremely important in the preparation of CMCs.

Ceramic particles are considered to be the best choice for reinforcing the Cu matrix, but ceramic particles may reduce the electrical conductivity of CMCs [11]. MAX phases are considered ideal ceramic reinforcing phases by researchers due to their high thermal and electrical conductivity, as well as machinability, compared to conventional binary carbide ceramic particles [1]. Among the MAX phase/Cu composite systems, the more common ones are Ti_3_AlC_2_/Cu, Ti2AlC/Cu, and Ti_2_AlN/Cu [11]. It has been shown that Ti_3_AlC_2_ and Cu react at elevated temperatures due to the unique structure and properties of MAX phase materials [3,12]. Zhang et al. indicated in their study that Al atoms of the A-layer can be de-embedded along the basal planes in Ti_3_AlC_2_/Cu composites, which induces the formation of Ti_x_C [3]. Solid solution (Ti_1-ε_Cu_ε_)_3_(Al,Cu)C_2_ is present in Ti_3_AlC_2_/Cu composites prepared by hot-press sintering, in which Cu substitutes for Al atoms in the A layer [13,14,15]. This is due to the fact that in the MAX phase, the A element is weakly bonded to M, whereas the covalent M–X bond is much stronger [16]. For Ti_3_AlC_2_, the submicron-thick TiCx layer forms a strong interfacial bond with the Cu(Al) alloy layer, which can achieve the purpose of effectively enhancing the mechanical properties of the Cu matrix [17]. As for the preparation of Cu matrix composites, powder metallurgy and the melt-casting method are the most commonly used processes. Compared with the powder metallurgy process, melt casting has the disadvantages of high energy consumption, environmental pollution, and the floating of reinforcing phase. Therefore, powder metallurgy is still the best choice for the preparation of copper matrix composites.

Earlier, Zhou et al. [18] reported that as the volume fraction of a MAX phase exceeds 20%, the densification of the composites decreases, thereby deteriorating the mechanical properties. The main reason is that the aggregation and distribution of MAX phase particles are not uniform, which leads to poor interfacial bonding between the matrix and the MAX phase. To address this problem, in this study, Cu-Ti_3_AlC_2_ composite powders were prepared via chemical copper plating, 20 Vol% Ti_3_AlC_2_/Cu and Cu@Ti_3_AlC_2_/Cu composites were prepared via vacuum hot pressing and sintering, and the microstructure and mechanical properties of the composites and powders were systematically evaluated.

## 2. Materials and Methods

### 2.1. Experimental Raw Materials

Copper powder (99.5% purity), Ti_3_AlC_2_ (99.5% purity, 400 mesh), a formaldehyde solution (99.5% purity, 0.10 mol/L), CuSO_4_·5H_2_O (99.0% purity), 2,2′-bipyridine (98.0% purity), potassium sodium tartrate tetrahydrate (99.0% purity), and disodium EDTA (99.0% purity) were used as experimental raw materials.

### 2.2. Preparation of Cu@Ti_3_AlC_2_ Powder

First, 10 g of CuSO_4_-5H_2_5O was dissolved in 300 mL of deionized water and magnetically stirred at room temperature for 50 min until all the CuSO_4_·5H_5_O was dissolved. Then, 3 g of EDTA-2Na with 6 g of potassium sodium tartrate with 0.6 g of polyethylene glycol was added to the solution configured in the previous step. The pH was adjusted to 13–14 using NaOH. An amount of Ti_3_AlC_2_ was added to the solution above with magnetic stirring. The formaldehyde solution was added dropwise to the mixed solution as a reducing agent for chemical copper plating, during which the NaOH solution was added continuously to maintain the pH of the chemical copper plating solution between 13 and 14. The copper plating reaction lasted for 50 min. The copper plating reaction lasted for 50 min, and the solid powder obtained by filtration was washed to neutrality with deionized water and dried under vacuum at 55 °C for 12 h to obtain copper-plated Ti_3_AlC_2_ powder (Cu@Ti_3_AlC_2_).

### 2.3. Preparation of Composite Material

The dried Cu@Ti_3_AlC_2_ powder was mixed with a certain amount of copper powder and ball milled at 400 rpm for 2 h. Anhydrous ethanol was used as the ball milling medium for the ball milling process, and argon was used as a protective gas during the ball milling time in order to protect the composite powder from oxidation. The Ti_3_AlC_2_/Cu composites were sintered in a vacuum hot-press sintering furnace, the powders were solidified into Ti_3_AlC_2_/Cu composites at a temperature of 850 °C and a pressure of 45 MPa, and the ordinary Ti_3_AlC_2_/Cu composites were prepared using the same conditions of ball milling and hot-press sintering as described above as a control material. The experimental procedure is shown in Figure 1.

### 2.4. Testing and Characterization

The physical structure analysis and microstructure characterization of the prepared composites and their powders were carried out using a scanning electron microscope (SEM), transmission electron microscope (TEM), X-ray diffraction (XRD), and energy spectrometer (EDS). The room temperature compression properties of the composites were investigated using a CMT5105 universal testing machine. The densities of the composites were evaluated using an AUY-120 densitometer.

## 3. Results and Discussion

### 3.1. Physical Phase Analysis and Microstructure Characterization of Composite Powders

The XRD pattern of the Cu@Ti_3_AlC_2_ powder, after 50 min of chemical copper plating and drying, is shown in Figure 2. Three diffraction peaks of Cu are observed in Figure 1, which correspond to the (111), (200), and (220) crystal planes of Cu, indicating that the divalent Cu ions were successfully reduced to Cu monomers during the chemical copper plating process and deposited on the surface of Ti_3_AlC_2_. The content of CuSO_4_-5H_2_O is 10 g, in which the content of Cu is 2.5 g, and the mass of the reinforced phase after chemical copper plating increased by 1.2 g. The content of Cu is very small compared with that of Ti_3_AlC_2_, so the Cu content is very small compared with that of Ti_3_AlC_2_. The content of Cu is very small compared to that of Ti_3_AlC_2_; therefore, the diffraction peaks of the three crystal surfaces of Cu are weak compared to the characteristic peaks of Ti_3_AlC_2_. In addition, the diffraction peaks shown in Figure 1 are all the diffraction peaks of Ti_3_AlC_2_ or Cu, and there are no other peaks; therefore, the MAX phase powders did not introduce other impurities during the copper plating process, and the structure of the MAX phase did not change significantly during the ball milling process. The XRD spectra of 20 Vol% Ti_3_AlC_2_/Cu composite powders after mixed ball milling are shown in Figure 3, and the Miller indices of the crystalline surfaces corresponding to the corresponding characteristic peaks are labeled, where (a) is the XRD spectra of Ti_3_AlC_2_ directly mixed with Cu, (b) is the XRD spectra of Cu@Ti_3_AlC_2_ mixed with Cu, and (c) is the XRD spectra of Cu@Ti_3_AlC_2_ powders, as shown in Figure 2. The positions of the characteristic peaks in Figure 3(a,b) are exactly the same, indicating that the composite powders have the same physical phase composition, i.e., they contain only Cu and Ti_3_AlC_2_. Comparing the curves in Figure 3(a–c), the peak intensities of Cu in the spectra (a) and (b) are higher than those in Figure 3(a–c), due to the fact that the volume fractions of Cu in the composite powders after ball milling of Ti_3_AlC_2_ or Cu@Ti_3_AlC_2_ with Cu are as high as 80%. Because the volume fraction of Cu in the composite powder after ball milling of Ti_3_AlC_2_ or Cu@Ti_3_AlC_2_ and Cu is as high as 80%, the peak intensity of Cu in the spectra (a) and (b) is much higher than that of Ti_3_AlC_2_, and the characteristic peaks of Cu and Ti_3_AlC_2_ are all in the same position, which indicates that ball milling for 2 h at 400 rpm did not distort the lattice of Cu and shifted the diffraction peaks and that this is just a physical process of uniformly mixing all kinds of powders to promote the diffusion of the elements in the sintering process [18].

Figure 4 demonstrates the microscopic morphology of the composite powders. As shown in Figure 4a, the ball-milled Cu and Ti_3_AlC_2_ particles showed irregular bars and flakes, and the size of the particles was about 10~30 μm. Figure 4b–d shows the morphology of the mixed ball-milled Cu@Ti_3_AlC_2_ and Cu powders under the low-resolution TEM, and the Cu has a larger atomic number, which presents a darker color under the electron microscope; the black area in the figure is the Cu coating on the Ti_3_AlC_2_ particles, and the lighter part is the Ti_3_AlC_2_ particles, which can be seen with the distribution of the Cu atoms composing the coating in the Ti_3_AlC_2_ particles. The black area in the figure is the Cu coating on the Ti_3_AlC_2_ particles, and the light-colored part is the Ti_3_AlC_2_ particles. It can be seen that the Cu atoms comprising the coating are distributed on the Ti_3_AlC_2_ particles and cover most of the area of the Ti_3_AlC_2_, but the thickness of the coating is not homogeneous, which may be due to the influence of the magnetic stirring. The obvious interface between Cu and Ti_3_AlC_2_ was observed in Figure 4c, and in the Cu plating from Figure 4d, it can be seen that the Cu and the copper-plated MAX phase powder were homogeneously mixed after ball milling, which is favorable for the homogeneous distribution of the Ti_3_AlC_2_ particles in the Cu substrate when performing the vacuum hot-press sintering. And the uniform dispersion of the reinforcing phase in the metal matrix is an important prerequisite for the realization of diffuse reinforcement [19,20]. Figure 4f shows the elemental distribution image of Cu@Ti_3_AlC_2_ composite powder in the region indicated in Figure 4e. It can be seen that out of the mixing ball milling, all the Cu, C, and Al elements are uniformly distributed. The Cu element shows a diffuse distribution in the composite powder while covering the surface of Ti_3_AlC_2_ particles. The uncovered enhanced phase particles, on the other hand, are detected with Ti elements, and this dispersion is consistent with the distribution presented by the low magnification TEM of the composite powder in Figure 4d. The microstructure of Cu@Ti_3_AlC_2_ powder was analyzed using HRTEM as shown in Figure 5. Figure 5b,c show the Fourier transform (FFT) images of region 1 and region 2, respectively, in HRTEM (Figure 5a). A set of crystalline surfaces in the Cu[-110] direction was detected in region 2, and Cu(1-1-1) crystalline surfaces were detected along with Ti_3_AlC_2_ (104) crystalline surfaces simultaneously in region 1. The inverse Fourier transform (IFFT) was measured to reveal that the Ti_3_AlC_2_ (104) facet has a crystallite spacing of about 0.211 nm and the Cu(1-1-1) facet has a crystallite spacing of about 0.229 nm. Dislocations due to energy transfer during ball milling were also detected in the IFFT image of the Ti_3_AlC_2_ (104) facet (Figure 5d).

### 3.2. Physical Phase Analysis and Microcharacterization of Composites

The XRD images of the Cu@Ti_3_AlC_2_/Cu composite (b) after vacuum hot-press sintering and the composite powder (a) before sintering are shown in Figure 6A,B, where B is the enlarged image of the area framed by the dashed line in A. The XRD images of the Cu@Ti_3_AlC_2_/Cu composite (a) after vacuum hot-press sintering are shown in Figure 6B is the enlarged image of the dashed area in A. It can be seen that the diffraction peaks of Cu and Ti_3_AlC_2_ after vacuum hot-pressing and sintering have both produced obvious shifts to small angles compared with those in the composite powder, and the peak intensities of Cu and Ti_3_AlC_2_ in (a) are obviously enhanced and sharper compared with those in (b), indicating enhanced crystallinity. The disappearance of the peak of Ti_3_AlC_2_ at 2θ = 19.22° indicates that the Al element diffuses from the A layer of the MAX phase into the Cu lattice to form a Cu(Al) solid solution, and a small amount of Cu also diffuses into the MAX phase [3], which leads to the enlarging of its lattice parameter. A small amount of TiC_x_ was also detected, which is due to the decomposition reaction of Ti_3_AlC_2_.
$${Ti}_{3}{AlC}_{2}+Cu\to Cu\left(Al\right)+TiC_{x}$$

Figure 7 shows the EDS spectral image of the Cu@Ti_3_AlC_2_/Cu composite, and the EDS results are consistent with the results of the XRD image; the mass ratio of Ti and Al elements is detected to be 11:1, and the atomic molar ratio is 3.6:1 instead of 3:1 in Ti_3_AlC_2_; again, this shows that the Al atoms diffuse into the lattice inside of the Cu and form a Cu(Al) solid solution, which is not easily detected. The formation of a solid solution triggers the accumulation of extended defects as well as microstrains, which can accelerate the densification of the composites during vacuum hot-press sintering, thus increasing the densification of the composites [21,22], and the TiC_x_ formed during this process forms a strong interfacial bond with the Cu(Al) alloy layer. This is the advantage that indicates that MAX phase materials are more suitable as a reinforcing phase for Cu-based composites compared to other ceramic particles.

Figure 8 demonstrates the microscopic morphology of the Cu@Ti_3_AlC_2_/Cu and Ti_3_AlC_2_/Cu composites, and the overall microstructure of the composites is quite homogeneous, with a more uniform distribution of the copper-plated Ti_3_AlC_2_ compared to the untreated one. In addition, as can be seen from the figures, although pores can be observed in Figure 8a,b, the number of pores is significantly lower than that in (c,d), and larger pores (red ellipses) as well as obvious interfacial gaps between Ti_3_AlC_2_ and Cu are observed in (c). In (a) and (b), there is a tight bond between Ti_3_AlC_2_ and Cu, with low porosity and no obvious interfacial gaps observed. The observed porosity is mainly due to the poor wettability between the MAX phase particles and the substrate and the direct contact of the reinforcer particles [23,24]. The wettability between the Ti_3_AlC_2_ particles and the Cu substrate is increased after the Cu-plating treatment, which reduces the probability of the Ti_3_AlC_2_ particles directly contacting each other and avoids the aggregation phenomenon to a certain extent [18]; the interfacial bonding effect is significantly stronger than that of the untreated Ti_3_AlC_2_, and this change leads to a significant enhancement of the mechanical properties of the composites.

The elemental distribution of Cu@Ti_3_AlC_2_/Cu composites is shown in Figure 9, where Ti_3_AlC_2_ is uniformly distributed in the Cu matrix, and it can be seen from the distribution of Ti elements (Figure 9c) that some of the Ti_3_AlC_2_ will still be agglomerated after the Cu-plating treatment, and the interfacial bonding between the agglomerated reinforcing phase and the matrix is poor. In addition, the Al element in the A layer behaves in a more dispersed manner in the Cu matrix compared with the Ti and C elements because of its unique diffusion behavior.

### 3.3. Mechanical Properties of Cu@Ti_3_AlC_2_/Cu Composites

Table 1 describes the densities and porosities of the Cu-based composites prepared before and after the copper plating of Ti_3_AlC_2_ particles. The densities of the Ti_3_AlC_2_ copper-plated composites are higher than those of the normal Ti_3_AlC_2_/Cu composites, with densities up to 7.66 g·cm^−3^, densities as high as 98.5%, and residual porosities of only 1.5%. This result is consistent with the SEM results that indicate that the interfacial wettability of Ti_3_AlC_2_ particles with Cu after surface copper plating treatment is increased, which enhances the bonding strength between Ti_3_AlC_2_ and matrix in the composites.

The compressive stress–strain curves of the 20 Vol% Ti_3_AlC_2_/Cu composites prepared in this study are shown in Figure 10. The maximum compressive strength of the composites prepared from the copper-plated 20 Vol% Ti_3_AlC_2_ particles reaches 705 MPa, and the compressive strain is 37.6, whereas the maximum compressive strength of the common 20 Vol% Ti_3_AlC_2_/Cu composites is only 651 MPa, and the compressive strain is 37.6. is only 651 MPa and the compressive strain is 33.5%. It can be seen that after the surface copper plating treatment of Ti_3_AlC_2_, the mechanical properties of the composites were significantly improved, the compressive strength increased by 8.29%, the compressive strain increased by 12.24%, and the ductility was also improved. This enhancement is well explained by the microstructure morphology in Figure 8 and the densification change results in Table 1. This increase in compressive strength and compressive strain are attributed to the reduced probability of agglomeration and the high interfacial bond strength between Ti_3_AlC_2_ and the Cu matrix after the surface copper plating treatment. The interfacial strength in composites determines if the reinforcing phase can effectively act as a load-bearing phase [25]. At the same time, the copper-plated reinforcing phase also has a stronger hindering effect on the dislocation motion during the deformation of the composites, which improves the compressive properties of the composites.

Figure 11 demonstrated the ultimate compressive strength and compressive strain of Ti_3_AlC_2_/Cu composites with different reinforcing phase contents in different studies. These include the 20 Vol% Ti_3_AlC_2_/Cu and Cu@Ti_3_AlC_2_/Cu composites prepared in this study as well as the Cu-based composites prepared by Huang et al. [16], using 50 Vol% Ti_3_AlC_2_ and Wang et al. [1], using 60 Vol% Ti_3_AlC_2_ and at different sintering temperatures. From the figure, it can be seen that the ultimate compressive strength of the composites was highest at 50 Vol% of Ti_3_AlC_2_ and a sintering temperature of 1050 °C, which reached 1185 MPa, but the compressive strain of the composites was only 2.9%. At 60 Vol% Ti_3_AlC_2_, the ultimate compressive strength reached 765 MPa, but the compressive strain was only 4.32%. The ultimate compressive strength of the 20 Vol% Cu@Ti_3_AlC_2_/Cu composite prepared in this study reached 705 MPa, and the compressive strain reached 37.6%. The large reduction in the compressive strain of the composites is attributed to the increase in weak bonding surfaces within the composites due to the increase in the content of the reinforcing phase, as well as the MAX phase ceramic particles, which are inherently very brittle. Zhang et al. [3] showed that the chemical reactivity between Ti_3_AlC_2_ and copper varies with increasing temperature. Their results showed that the reactivity was lower in the range of 850–950 °C and higher above 950 °C. In this study, only a small amount of TiC_x_ was produced at a sintering temperature of 850 °C. In this study, only a small amount of TiC_x_ was generated at a sintering temperature of 850 °C. The structural integrity of Ti_3_AlC_2_ was maintained to a certain extent, and the bonding surface between Ti_3_AlC_2_ and the matrix was improved along with copper plating, resulting in a good balance of compressive strength and ductility of the composite.

## 4. Conclusions

Cu@Ti_3_AlC_2_ powder was prepared via copper plating on the surface of Ti_3_AlC_2_ particles using chemical plating, and Cu-based composites were prepared by vacuum hot-press sintering after mixing with Cu powder. The Cu@Ti_3_AlC_2_ powders were analyzed and characterized by SEM, TEM, and XRD after the copper plating treatment. The particle size of the powders was about 10~30 μm, and the presence of Cu was successfully detected on the surface of Ti_3_AlC_2_; no other impurities were found. Compared with ordinary Ti_3_AlC_2_/Cu composites, the Cu@Ti_3_AlC_2_/Cu composites have higher densities and interfacial bonding strengths and avoid direct contact with Ti_3_AlC_2_, which reduces the probability of agglomeration phenomena to a certain extent and leads to a more homogeneous dispersion of the reinforcing phase. The maximum compressive strength reached 705 MPa, which was 8.29% higher than that of the Ti_3_AlC_2_/Cu composites, and the compressive strain reached 37.6%, which was 12.24% higher than that of the Ti_3_AlC_2_/Cu composites.

## Figures and Tables

**Figure 1 nanomaterials-14-00418-f001:**
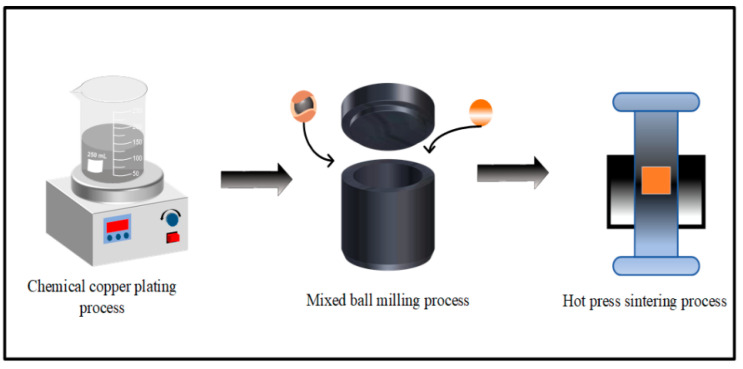
Experimental flow chart.

**Figure 2 nanomaterials-14-00418-f002:**
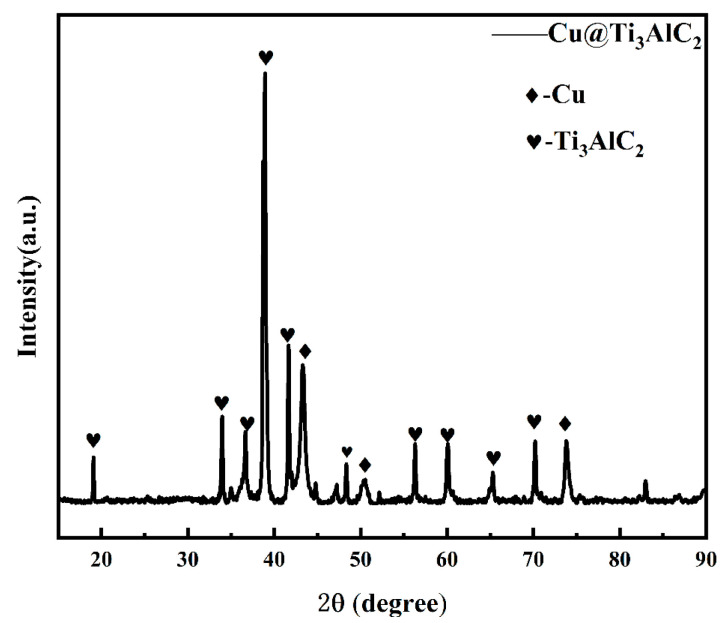
XRD pattern of Cu@Ti_3_AlC_2_.

**Figure 3 nanomaterials-14-00418-f003:**
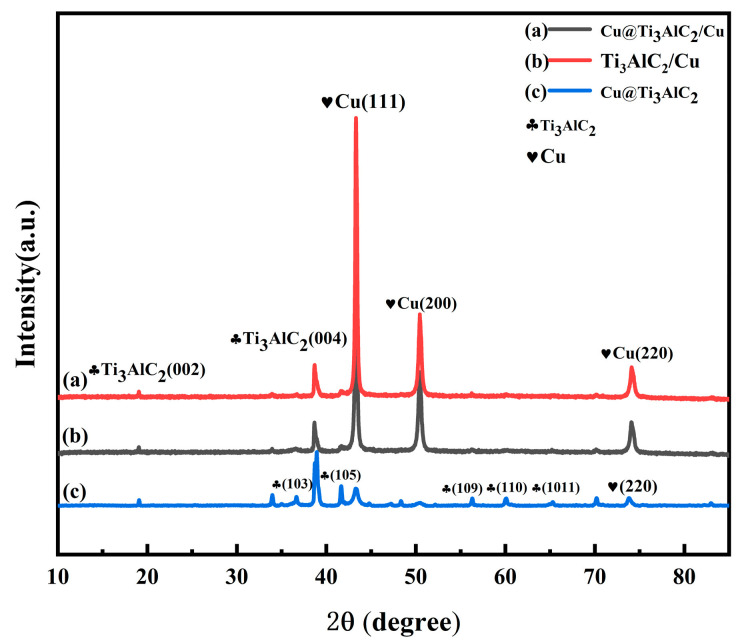
XRD pattern of 20 Vol% Ti_3_AlC_2_/Cu composite powder.

**Figure 4 nanomaterials-14-00418-f004:**
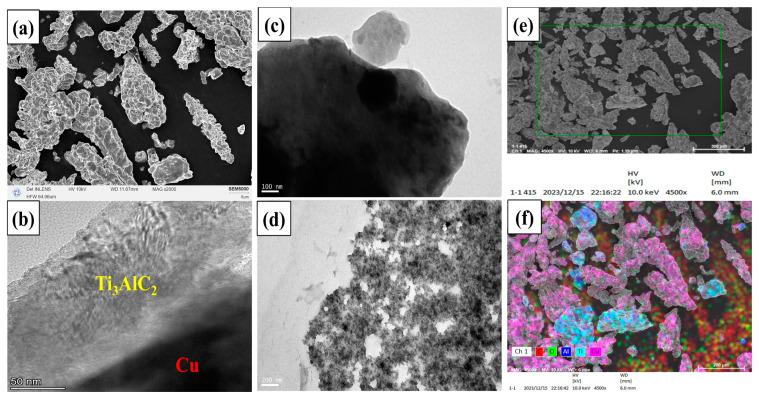
(**a**–**f**) Microscopic morphology of Cu@Ti_3_AlC_2_ powders and elemental distribution of composite powders.

**Figure 5 nanomaterials-14-00418-f005:**
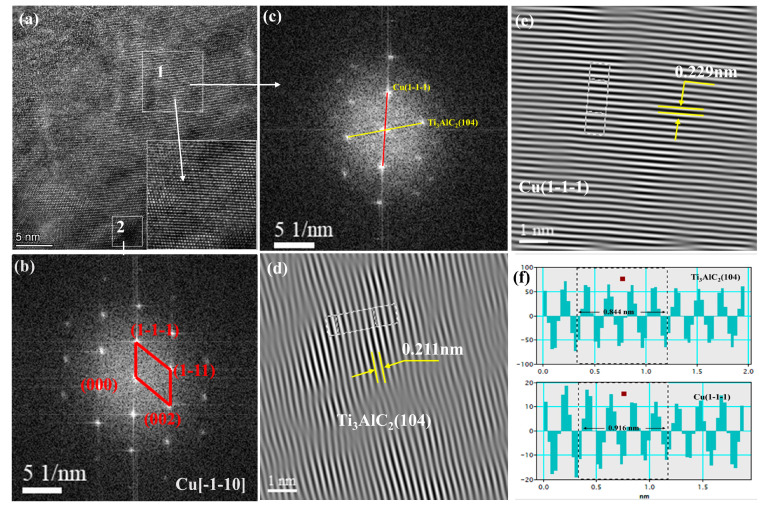
Interfacial structure of Cu@Ti_3_AlC_2_: (**a**) HRTEM image of Cu@Ti_3_AlC_2_; (**b**) FFT image of region 2; (**c**) FFT image of region 1; (**d**–**f**) IFFT images of Ti_3_AlC_2_ vs. Cu with measurements of crystalline spacing.

**Figure 6 nanomaterials-14-00418-f006:**
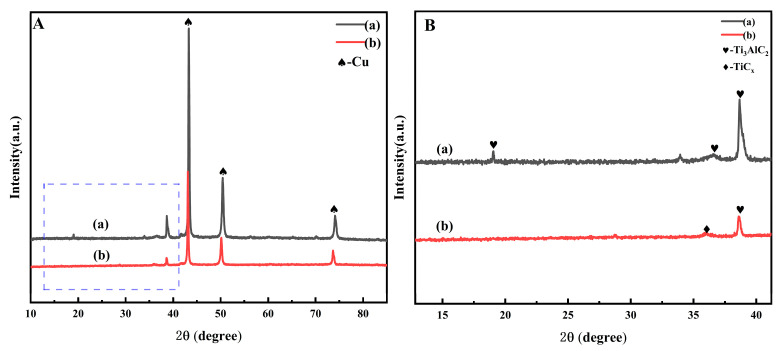
(**A**) XRD comparison of 20 Vol%Cu@Ti_3_AlC_2_/Cu composite powder and composite material; (**B**) localized area magnification; (**a**): composite powder; (**b**): composite material.

**Figure 7 nanomaterials-14-00418-f007:**
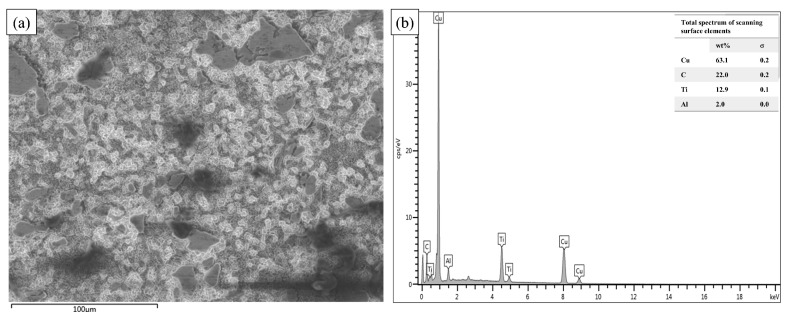
(**a**) Organizational morphology of Cu@Ti_3_AlC_2_/Cu composites; (**b**) corresponding EDS energy spectra.

**Figure 8 nanomaterials-14-00418-f008:**
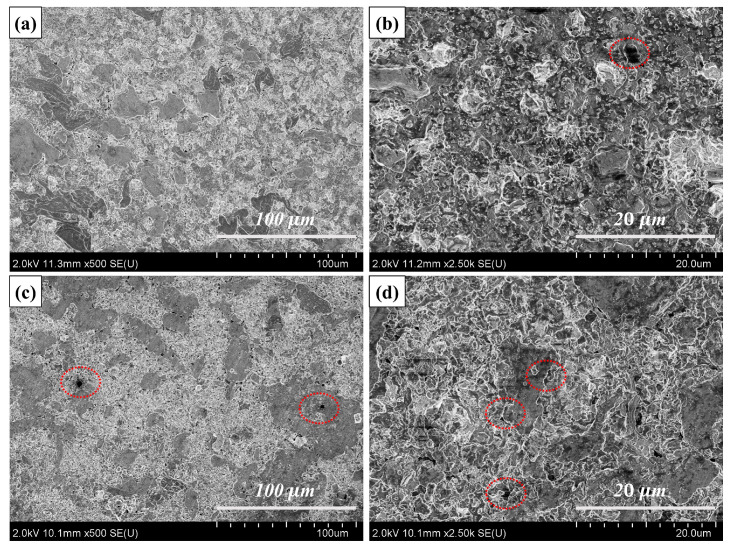
SEM images of 20 Vol% Ti_3_AlC_2_/Cu composites: (**a**,**b**) Cu@Ti3AlC2/Cu; (**c**,**d**) Ti_3_AlC_2_/Cu.

**Figure 9 nanomaterials-14-00418-f009:**
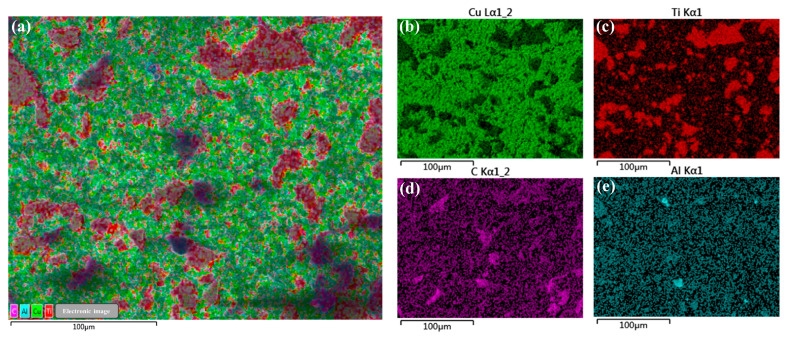
(**a**–**e**) Elemental distribution of Cu@Ti_3_AlC_2_/Cu composites. (**a**) Cu@Ti_3_AlC_2_/Cu element distribution map; (**b**) Distribution of Cu elements; (**c**) Distribution of Ti elements; (**d**) Distribution of the C element; (**e**) Distribution of Al elements.

**Figure 10 nanomaterials-14-00418-f010:**
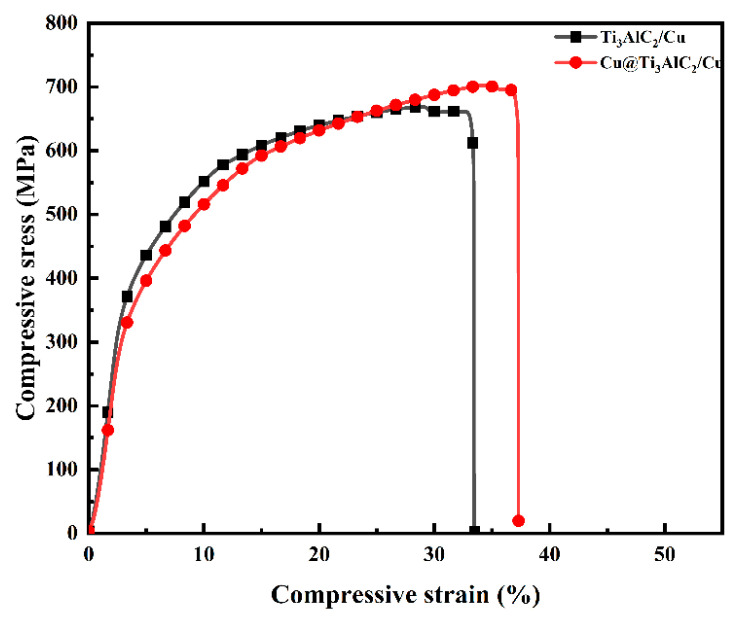
Compressive stress–strain curves of 20 Vol% Ti_3_AlC_2_/Cu composites.

**Figure 11 nanomaterials-14-00418-f011:**
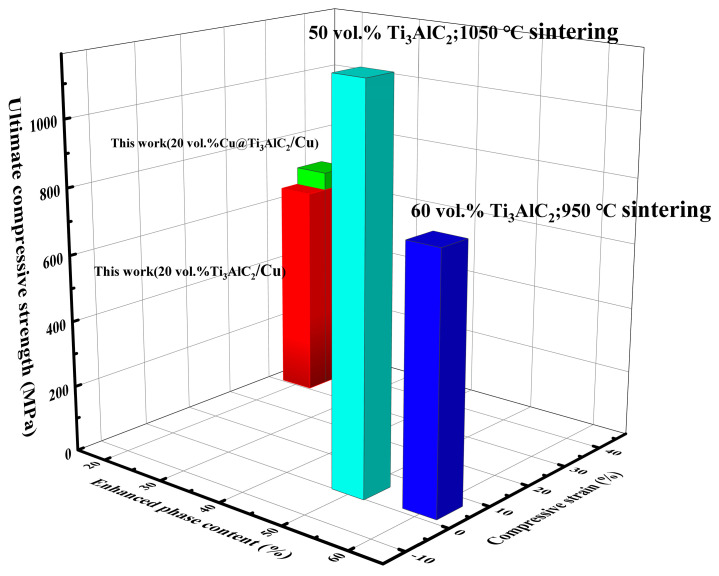
Mechanical properties of Ti_3_AlC_2_/Cu composites under different studies.

**Table 1 nanomaterials-14-00418-t001:** Densities, compactness, and porosity of Ti_3_AlC_2_/Cu composites.

Composite Material	Theoretical Density (g·cm^−3^)	Density (g·cm^−3^)	Compactness (%)	Porosity (%)
Cu@Ti_3_AlC_2_/Cu	7.77	7.66	98.5	1.5
Ti_3_AlC_2_/Cu	7.77	7.53	96.9	3.1

## Data Availability

Data are contained within the article.

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
