# Peer review of "Mechanical Properties of Ti3AlC2/Cu Composites Reinforced by MAX Phase Chemical Copper Plating"

_nanomaterials, 2024, doi:10.3390/nano14050418_

Round 1

Reviewer 1 Report

Comments and Suggestions for Authors

The research article: Mechanical properties of Ti3AlC2/Cu composites reinforced by MAX-phase chemical copper plating

This research synthesised Ti3AlC2/Cu composites by MAX-phase chemical copper plating with the following characterisation and mechanical properties testing. Test results reveal that the density of the Cu@Ti3AlC2/Cu composite reaches 98.5%, with a maximum compressive strength of 705 MPa, which is 8.29% higher than that of the Ti3AlC2/Cu composite.

1.     Figures 2 and 3, XRD patterns should be added to one graph. Moreover, the XRD pattern of Ti3AlC2 should be added. ICCD patterns for compounds and the Miller indexes added.

2.     Figure 5. All XRD patterns should be shown simultaneously with Figure 2 and Figure 3.

3.     The Raman spectroscopy employment in this study would be welcome. It is a more sensitive technique for internal structure characterisation than XRD.

4.     The images are too small. In most cases, there is room for an increase in image size.

The manuscript requires a major revision.

Comments on the Quality of English Language

Language should be double-checked.

Author Response

Dear reviewer:
Your questions for this article are greatly appreciated.
As for Figure 2 and Figure 3, I have integrated the XRD data and presented them in the paper, and added the Miller index for your review.
With regard to Raman spectroscopy, we deeply regret that we were not able to use Raman technology to characterize the material during the Spring Festival. XRD, TEM, SEM and EDS were used to characterize the composite powder and composite materials
As for the image problem, I have enlarged the image in the article appropriately, please review.
Finally, we would like to thank you again for your advice

Reviewer 2 Report

Comments and Suggestions for Authors

The article under review presents a comprehensive study on the mechanical properties of Ti3AlC2/Cu composites reinforced by MAX-phase chemical copper plating. It outlines the preparation process of Cu-coated Ti3AlC2 particles through electroless plating and their subsequent incorporation into Cu matrices via vacuum hot pressing sintering. Through various characterization techniques, including XRD, SEM, and TEM, the article demonstrates successful copper plating on Ti3AlC2 particles and the improved mechanical properties of the resulting composites. Overall, the study presented in the article provides significant insights into the use of MAX phases as reinforcing materials in metal matrix composites. Here are some suggestions:

1.      The article mentions the high electrical conductivity of MAX phases as a significant advantage for their use as ceramic reinforcing phases. However, it lacks data to support this claim in the context of the formed composites. Providing electrical conductivity measurements before and after forming the composite with MAX would offer a more comprehensive understanding of the material's performance and validate the theoretical advantages discussed.

2.      Given the importance of understanding the distribution of copper within the composite, adding EDX mapping for either figure 4c or 4d would greatly enhance the clarity of Cu's distribution.

3.      The final chemical composition of the composite is crucial for evaluating the success of the fabrication process. Including detailed chemical composition based on techniques such as ICP would provide valuable information on the elemental distribution within the composite. 

Author Response

Dear reviewer:
Your questions for this article are greatly appreciated.
Thank you very much for your question about the conductivity. Although conductivity is the advantage of the MAX phase, this article focuses on the layered structure of the MAX phase and the easy diffusion of the A layer to enhance the mechanical properties. This study also focuses on the effect of Cu coating on the mechanical properties of the composite.
Regarding the distribution of elements in Figure 4c or d, I have added the element distribution images of composite powder in the paper as Figures 4e and 4f for your review.
The problem of the final chemical composition of the composite is presented using XRD patterns and element distribution images. It is a pity that we were not able to use ICP and other technologies for analysis during the Spring Festival.
Thank you again for your valuable advice

Round 2

Reviewer 1 Report

Comments and Suggestions for Authors

Dear Authors,

thank you for the significant improvements to the manuscript.